# Mitochondrial Quality Control in the Heart: The Balance between Physiological and Pathological Stress

**DOI:** 10.3390/biomedicines10061375

**Published:** 2022-06-10

**Authors:** Giovanni Fajardo, Michael Coronado, Melia Matthews, Daniel Bernstein

**Affiliations:** 1Department of Pediatrics and the Cardiovascular Institute, Stanford University, Stanford, CA 94305, USA; gfajardo@stanford.edu; 2Cytokinetics Inc., South San Francisco, CA 94080, USA; mcoronado@cytokinetics.com; 3Department of Biomedical and Biological Sciences, Cornell University, Ithaca, NY 14850, USA; mdm345@cornell.edu

**Keywords:** mitochondria, fission, fusion, mitophagy, biogenesis

## Abstract

Alterations in mitochondrial function and morphology are critical adaptations to cardiovascular stress, working in concert in an attempt to restore organelle-level and cellular-level homeostasis. Processes that alter mitochondrial morphology include fission, fusion, mitophagy, and biogenesis, and these interact to maintain mitochondrial quality control. Not all cardiovascular stress is pathologic (e.g., ischemia, pressure overload, cardiotoxins), despite a wealth of studies to this effect. Physiological stress, such as that induced by aerobic exercise, can induce morphologic adaptations that share many common pathways with pathological stress, but in this case result in improved mitochondrial health. Developing a better understanding of the mechanisms underlying alterations in mitochondrial quality control under diverse cardiovascular stressors will aid in the development of pharmacologic interventions aimed at restoring cellular homeostasis.

## 1. Energy Homeostasis

The energetic homeostasis of living organisms is maintained by balancing fuel inputs, energy production, energy use, and waste products. Energetic stress occurs when there is disruption of any component that maintains homeostasis. Although many factors can cause energetic stress, among the most disruptive are imbalances in energy production, including metabolic dysfunction, fuel deficit, and oxygen deprivation [1].

Energetic stress can be either physiological or pathological in nature, however, both types share many common downstream effects. Physiological stimuli, such as exercise, usually result in a beneficial or adaptive response, e.g., increased mitochondrial function in skeletal muscle or enhanced contractility in the heart. In comparison, pathological stimuli like trauma or disease can lead to negative responses, from cellular apoptosis to full organ shutdown. In both contexts, the energy balance is challenged, but only with disease do we see negative long-term consequences. Indeed, repeat bouts of physiologic stress can result in protection against pathologic stress, e.g., the benefits of regular aerobic exercise or ischemic pre-conditioning.

## 2. Mitochondrial Dynamics Maintain Energy Balance

Mitochondria are both creators and regulators of metabolic energy stores. Mitochondria maintain an efficient and productive population by undergoing physical changes in their structure through the processes of fusion, fission, and mitophagy. Their dynamic morphology allows mitochondria to play a role in many aspects of cellular homeostasis including quality control, metabolic energy supply, cell signaling, apoptosis and aging [2]. The constant fission, fusion, and turnover of these organelles provides avenues for a cellular response to ever-changing physiological and pathological conditions (Figure 1).

Fusion, the act of combining two or more mitochondria into one, is mediated by both outer mitochondrial membrane proteins (mitofusin 1 (MFN1) and mitofusin 2 (MFN2)) and inner membrane proteins (optic atrophy factor 1 (OPA1)) [5]. Fusion requires energy, through the hydrolysis of GTP. An integral role of fusion is to allow a mixing of cellular content between two different mitochondria, including proteins, metabolites, and mitochondrial DNA.

Fission is the act of dividing one mitochondrion into two or more mitochondria. Dynamin-related protein 1 (Drp1) is a critical mediator of fission that binds to the mitochondrial surface proteins human fission factor 1 (FIS1), mitochondrial dynamics proteins 49 and 51 (MiD49/51), and mitochondrial fission factor (MFF). Once Drp1 is bound to mitochondria, it initiates contraction of the membrane through GTP hydrolysis. Fission plays several roles, including: (1) activation of mitophagy, (2) increasing the number and surface area of mitochondria, and (3) allowing for easier movement of mitochondria to different parts of the cell, important for cell division in many organs, but not in the heart. Fission is essential for cellular and mitochondrial homeostasis by contributing to improved mitochondrial quality control. This can occur through asymmetric fission with the segregation of functional from dysfunctional mitochondrial components into different daughter mitochondria, and the subsequent elimination of the resultant dysfunctional mitochondria through mitophagy [6]. Although asymmetric fission is not a required prerequisite for mitophagy, it does play an important role in enriching healthier mitochondrial populations.

Mitophagy is the process for the removal of damaged mitochondria from the cellular population. Activation of mitophagy can occur through multiple pathways that are Parkin-dependent and independent. Parkin-dependent mitophagy involves the mitochondrial recruitment of Parkin, p62, and PTEN-induced putative kinase 1 (PINK1) [2]. The process begins with damaged mitochondrial signaling to recruit PINK1 to the outer mitochondrial membrane. This allows the stable binding of Parkin, which then induces ubiquitination of MFN1, halting fusion and recruiting adaptor proteins such as p62 to signal the process of autophagy [7]. While the PINK1/Parkin pathway is one of the most canonical mitophagy pipelines, there are other Parkin-independent pathways that produce the same result of improved mitochondrial quality control [8]. Outer mitochondrial membrane proteins can be ubiquitinated by E3 ligases other than Parkin, and OMM mitophagy receptors can interact directly with components of the autophagosomal membrane such as LC3-II and GABARAP. PINK1 can also recruit adaptor proteins such as NDP52 and optineurin to the mitochondria. These factors play an important role in recruiting and activating autophagy regulators ULK1, DFCP1 and WIPI1 to the mitochondria, resulting in LC3-mediated mitophagy [9]. Finally, proteins that are anchored in the mitochondrial membranes can also function as mitochondrial receptors for autophagosomes. The BCL-2-related proteins BNIP3 and BNIP3L/NIX as well as FUN14 domain-containing protein 1 (FUNDC1), a highly conserved outer mitochondrial membrane protein, act as mitophagy receptors that can bind directly to LC3 to clear mitochondria, eliminating the need for adaptor proteins [10]. FUNDC1 also interacts with both DRP1 and OPA1 to coordinate mitochondrial fission or fusion and mitophagy [11]. Thus, multiple pathways exist to activate mitophagy and maintain mitochondrial quality control.

The normal function of the mitochondrial dynamics machinery is integral to the maintenance of homeostasis and thus disruption of this machinery can precipitate disease [12]. The dynamic nature of mitochondria allows them to rapidly respond to changes in energetic demand. However, some tissues are more susceptible to energetic stress due to their high energy needs and the degree to which altered dynamics plays a role differs between different tissues. The heart is an example of where normal hemodynamic function requires large and sustained energy production. Even a brief period of hypoxia or ischemia can result in significant damage, thus, unique mitochondrial adaptations to stress have developed in the mammalian heart.

## 3. Cardiac Mitochondria Respond to Energetic Stress

In order to meet energy needs, cardiac mitochondria make up ~30% of total cell volume, creating over 6 kg of ATP each day (200 tons of ATP during a normal lifetime) through oxidative phosphorylation [13]. In order to maintain energy homeostasis during fluctuating energetic needs (e.g., alternating periods of rest vs. exercise), the heart must respond dynamically to modulate mitochondrial function. In response to increased physiological demand, such as occurs during aerobic exercise, the sympathetic nervous system releases adrenergic neurotransmitters that initiate a multipronged response to enhance heart rate and cardiac contractility. In cardiomyocytes, sympathetic signaling increases calcium flux from both external and sarcoplasmic reticulum stores, shifts the balance of myosin heads from the super-relaxed state to the active state, and enhances myosin crossbridge cycling, resulting in enhanced heart rate and contractility, thereby increasing cardiac output [14]. In the vasculature, sympathetic signaling combines with metabolic autoregulation to direct blood flow to working skeletal muscles through targeted vasodilation, and away from other vascular beds through targeted vasoconstriction. Overall these changes result in enhanced venous return, ventricular end-diastolic volume and pressure, and cardiac output due to the Frank–Starling effect [15].

A consequence of enhanced heart rate and contractility is the need for more energy [16]. Recent studies have shown that the sympathetic response also plays an important role in addressing increasing energy needs by modulating mitochondrial function, dynamics and fuel utilization. The stimulation of adrenergic receptors in the liver and adipose tissues activates glycogenolysis and lipolysis, respectively, resulting in elevated circulating fuel sources such as glucose and free fatty acids [17,18]. Individual cells stimulated by adrenergic receptors can also activate glycogenolysis and lipolysis, increasing endogenous fuel sources for energy production [17,18]. The same sympathetic simulation can also enhance mitochondrial function. The activation of beta-adrenergic receptors with isoproterenol in mouse HL-1 cardiomyocytes results in enhanced basal and maximal oxygen consumption. This enhanced respiration mediated by sympathetic stimulation is also associated with rapid changes in mitochondrial morphology, resulting in mitochondria that are smaller and more spherical in shape [19]. If either beta-adrenergic receptor signaling or mitochondrial fission pathways are inhibited during sympathetic stimulation, enhanced mitochondrial function is blocked [19]. These findings suggest an elegant regulatory system, where adaptions to hemodynamic and energetic demands occur through parallel processes, mediated in large part by the sympathetic nervous system.

The initial phases of many physiological and pathological stressors induce similar responses in order to maintain bioenergetic homeostasis, where sudden changes in energetic demands or supply must be met by a mitochondrial response. However, differences begin to arise during the resolution phase of the response to physiological vs. pathological stress, resulting in either reestablished homeostasis or chronic dysfunction.

## 4. Physiological Adaptations

### 4.1. Acute Exercise

Aerobic exercise is a common physiological stressor during which the body must rapidly increase hemodynamic function and energy production to meet the demands of exercise on both skeletal and cardiac muscle. The hemodynamic response to acute exercise occurs rapidly, with an immediate rise in heart rate (HR) due to reduced parasympathetic tone followed by activation of the sympathetic nervous system via β-adrenergic receptors [20]. This rapid rise in HR and contractility requires an equally rapid increase in mitochondrial function to sustain the high ATP demands of the exercise-stressed heart [16]. Since adaptations to acute exercise must occur immediately, transcriptional- and translational-dependent mechanisms like mitochondrial biogenesis cannot contribute to these acute energetic responses. Instead, the body relies on more rapid mechanisms that can directly enhance mitochondrial respiratory function and maximize mitochondrial surface area. At the onset of exercise, β-adrenergic signaling results in enhanced cytosolic Ca^2+^ via activation of the sarcoplasmic reticulum ryanodine receptor (RyR) and sarcolemmal L-type Ca^2+^ channels, resulting in an enhanced inward Ca^2+^ current [21]. Increased cytosolic Ca^2+^ serves to directly activate myosin-actin cross bridge cycling at the sarcomere, and in parallel to activate mitochondrial enzyme function by entering the mitochondria through the mitochondrial calcium uniporter (MCU). Elevated mitochondrial Ca^2+^ enhances ATPase activity, dehydrogenase activity, and NADH oxidation, resulting in increased ATP production [22,23]. Interestingly, the expression and activity of the MCU, and of Ca^2+^ import to the mitochondria, is itself dependent on β-adrenergic signaling and levels of cytosolic Ca^2+^ [24]. Thus, Ca^2+^ acts as a potent and acute response mediator to changes in energetic demand that can rapidly enhance mitochondrial function through the modulation of enzymatic activity.

Acute cardiac energetic adaptations to exercise also occur through changes in mitochondrial morphology, specifically through activation of mitochondrial fission. Traditionally, mitochondrial fission has been associated with pathological conditions such as ischemia. However, in the case of exercise, mitochondrial fission plays a critical role, and is actually required to achieve maximal exercise [19]. Ca^2+^ again plays an important regulatory role in modulating mitochondrial morphology, where elevated cytosolic Ca^2+^ activates the Ca^2+^-dependent enzymes calcineurin and CamKII, resulting in the dephosphorylation of inhibitory phosphate Ser637 and the phosphorylation of activating phosphate Ser616 on fission mediator Drp1 [25,26], respectively. In parallel, the downstream effectors protein kinase A (PKA) and AMP-activated protein kinase (AMPK) also phosphorylate Drp1 at Ser 637 and Ser616, respectively, resulting in Drp1 activation [27,28]. Thus, several signaling pathways converge to activate mitochondrial fission as an adaptive response to increased energetic demands. Cardiac mitochondria isolated from exercised mice tend to be smaller in size, more circular, and have increased state 3 oxygen consumption rate, whereas knockout of cardiac Drp1 results in reduced maximal and submaximal exercise capacity and a faster time to anaerobic threshold [19]. Thus, mitochondrial fission is a potent acute physiological response to increased energetic demands that can rapidly enhance mitochondrial function. We have labelled this process “physiologic fission” to distinguish it from the “pathologic fission” that occurs after stresses such as ischemia [19].

### 4.2. Chronic Exercise

Chronic aerobic exercise, such as occurs with endurance training, improves health outcomes and enhances cardiac function. Physiological eccentric hypertrophy, reduced systemic vascular resistance, and reduced resting heart rate are all associated with endurance exercise training and allow for the greater use of passive forces, like preload, to generate force during contraction [29]. Mitochondrial biogenesis, the generation of new mitochondria, is another key adaptive mechanism that occurs with exercise training, where changes in gene expression lead to an increase in mitochondrial mass [30]. One specific response to moderate endurance exercise is the upregulation of nitric oxide synthase (NOS), which increases mitochondrial biogenesis in cardiac tissues [31]. PCG-1α, a coactivator of transcription factors and a synchronizer of the mitochondrial and nuclear genome, is also upregulated in situations with increased energetic stress such as chronic endurance exercise [30,32]. PGC-1α acts as a regulator for mitochondrial biogenesis and metabolism, controlling the number and respiratory function of mitochondria in response to variations in energy demands [33]. The expansion of mitochondrial mass that results from PGC-1α and NOS activation can increase the energy-producing capacity of the heart and allows for greater sustained aerobic energy production [31,34].

In addition to the expansion of mitochondrial mass, exercise training can also improve mitochondrial quality through repeated cycles of fission and mitophagy. Mitochondrial DNA (mDNA) is highly susceptible to ROS-mediated damage due to the low capacity of antioxidants and mDNA repair enzymes in the mitochondrial matrix [35]. Since the mitochondria are the major producers of endogenously derived ROS, mitochondria tend to have a high mDNA mutation rate, which can result in misfolded and dysfunctional mitochondrial proteins and reduced mitochondrial function [35]. Endurance exercise training can help sustain or even improve mitochondrial quality by activating repeated bouts of mitochondrial fission and mitophagy, resulting in the asymmetric segregation and elimination of dysfunctional mitochondria [36,37]. Through this mechanism, fission plays an important role in separating dysfunctional mitochondria from healthier populations [34]. Dysfunctional mitochondria, which tend to have a lower membrane potential, are targeted by membrane potential-sensitive enzymes like Pink1 and Parkin, resulting in ubiquitination and destruction by autophagosomes [37]. Multiple studies have demonstrated that endurance exercise can induce mitophagy during post-exercise rest by the activation of fission and mitophagy pathways [38,39,40]. More recently, investigators have demonstrated that AMPK physically localizes to the outer mitochondrial membrane during exercise to act as a metabolic sensor of energetic stress and to facilitate mitophagy [41]. Thus, chronic endurance exercise can improve mitochondrial quality by activating first fission to segregate and then mitophagy to destroy dysfunctional mitochondria.

## 5. Pathological Adaptations

Adaptations to pathologic energetic stress involve many interweaving pathways that trigger a cascade of responses that range from changes in Ca^2+^ flux to widespread remodeling of mitochondrial morphology and mass. As mentioned above, many of the same adaptations and signaling events that occur with physiological stressors, like exercise, also occur during pathological stressors such as cardiovascular disease. In the case of pathological stressors and adaptation, mitochondrial quality control is fundamental for the removal of defective mitochondria and the replenishment of the mitochondrial network [42,43]. Alterations in mitochondrial fission, fusion, mitophagy and biogenesis play a role in multiple cardiovascular diseases including ischemia/reperfusion injury [43,44]; ischemic [45,46], diabetic [47,48], anthracycline [49,50] and septic cardiomyopathies [51]; pressure-overload hypertrophy [52]; heart failure [53]; stroke [54]; and myocarditis [55], among others. More comprehensive reviews of the mechanistic insights of alterations in mitochondrial dynamics and quality control in cardiovascular diseases have been published [12,56,57,58,59,60,61]. Here we summarize some of the mitochondrial dynamics and quality control alterations described in the most prevalent cardiovascular diseases (Table 1).

### 5.1. Ischemic Heart Disease

Mitophagy is protective during and after ischemia since it helps to remove abnormal mitochondria. Parkin-deficient mice at baseline show normal cardiac function despite disorganized mitochondrial networks and significantly smaller mitochondria. However, these mice are more sensitive to myocardial infarction. Parkin knockout myocytes had reduced mitophagy and accumulated swollen, dysfunctional mitochondria after infarction; in addition, an overexpression of Parkin in isolated cardiomyocytes protected against hypoxia-mediated cell death, suggesting that mitophagy plays a protective role against ischemia [45]. It has been shown that mitophagy mediated through the Ulk1/Rab9/Rip1/Drp1 pathway protects the heart against ischemia by maintaining healthy mitochondria [78]. In addition, PINK1 knockout mice developed significantly larger myocardial infarcts following ischemia-reperfusion injury; an overexpression of PINK1 in HL-1 cells delayed the time to mitochondrial permeability transition pore opening and reduced cell death following simulated ischemia/reperfusion (IR) [62]. Furthermore, Opa1 overexpression protected cardiomyocytes against hypoxia-induced damage and enhanced cell viability by inducing mitophagy [63]. Pigment epithelial-derived factor (PEDF) protects against hypoxia-induced apoptosis and necroptosis in cardiomyocytes; PEDF promotes FUNDC1-mediated cardiomyocyte mitophagy via ULK1 [64]. Thus, OPA1-induced mitophagy and FUNDC1-dependent mitophagy can be cardioprotective against ischemia [79].

However, this adaptive process must be carefully regulated as excessive mitophagy alters mitochondrial integrity and function, which can be harmful to cell survival, especially during reperfusion [60]. Post-ischemic G protein-coupled estrogen receptor 1 activation induces cardioprotective effects against ischemia/reperfusion injury by protecting mitochondrial structural integrity and function and reducing mitophagy [65]. The activation of ALDH2 with Alda-1 protects cardiomyocytes against IR injury by suppressing PINK1/Parkin-dependent mitophagy via reducing 4HNE accumulation [66]. Furthermore, compared to WT, Akap1 knockout mice displayed larger infarct size, decreased cardiac function and survival associated with mitochondrial structural abnormalities, reduced mitochondrial function, and enhanced cardiac mitophagy and apoptosis [67].

Excessive mitochondrial fission occurs within 60 min of reperfusion after transient myocardial ischemia, leading to mitochondrial dysfunction and decreasing cardiac contractility [80]. In both isolated neonatal murine cardiomyocytes and adult rat hearts, mitochondrial fragmentation and swelling occur within 30 min of ischemia-reperfusion. Drp1-Serine 637 dephosphorylation results in Drp1 mitochondrial translocation and increased fission as well as cytosolic calcium overload, and it promotes cardiomyocyte death and myocardial contractile dysfunction [58,68].

Although mitochondrial fusion tends to be protective under physiological conditions, the role of fusion mediators in ischemia-reperfusion remains controversial [43]. Promoting mitochondrial fusion has been shown to exert cardioprotection against ischemia-reperfusion injury through an increase in mitochondrial fusion events, ultimately leading to attenuated cardiac arrhythmia, reduced infarct size, and improved LV function [56,69]. OPA1, mitochondrial fusion, and mitophagy were significantly repressed by ischemia-reperfusion injury, accompanied by an expansion of the infarct area and worse cardiac function [81]. Low-level OPA1 overexpression protected from ischemia-reperfusion, improving enzymatic parameters associated with tissue damage in response to ischemic injury, suggesting that cristae remodeling is also central to propagating necrotic damage, and that its prevention can ameliorate ischemic injury outcomes [70]. However, Mfn-2 knockout mice develop higher pressures during post ischemic reperfusion and exhibit diminished cell death following in vivo regional ischemia and reperfusion injury, suggesting that Mfn-2 not only serves to maintain mitochondrial morphology in cardiac myocytes but also promotes mitochondrial permeability transition pore opening in the heart under conditions of stress [82].

### 5.2. Cardiac Hypertrophy and Pressure Overload-Induced Heart Failure

The activation of mitophagy is critical for maintaining mitochondrial function and cardiac performance in the presence of pressure overload. Mitophagy is transiently activated at 3 to 7 days after transverse aortic constriction in mice, coinciding with mitochondrial translocation of Drp1; however, it is downregulated thereafter, followed by mitochondrial dysfunction [71]. The transient upregulation of mitophagy by TAC is mediated through an Atg7-dependent mechanism, peaking at 1 day, followed by a stronger activation (with a delayed time course) through an Ulk1-dependent mechanism. Ulk1-mediated alternative mitophagy is a major mechanism in response to pressure overload and plays an important role in mediating mitochondrial quality control mechanisms and protecting the heart against cardiac dysfunction in response to increased afterload [83]. In cardiomyocytes, hypertrophy induced by the stimulation of α1-adrenergic receptors with norepinephrine increases cytoplasmic calcium, activating calcineurin and Drp1 translocation to mitochondria, thus inducing fission, and is associated with a decrease in mitochondrial function. Dominant-negative Drp1 prevented fission and blocked norepinephrine-induced hypertrophy [72].

Altered mitochondrial biogenesis is one of the underlying mechanisms of mitochondrial dysfunction in adverse cardiac remodeling. A down-regulation of genes involved in energy metabolism modulation and mitochondrial biogenesis has been shown in several experimental models of heart failure [84]. Pressure overload cardiac hypertrophy and heart failure are associated with impaired mitochondrial function that correlates with the repression of PGC-1α expression [73]. However, in human heart failure with preserved ejection fraction (HFpEF), there is mitochondrial fragmentation, cristae destruction, and vacuolar degeneration without changes in the expression of mitochondrial biogenesis and dynamic markers, except for an increase in BNIP3 expression and its dimerization [85]. PINK1 protein levels are markedly reduced in end-stage human heart failure. In addition, PINK1 knockout mice develop left ventricular dysfunction and evidence of pathological cardiac hypertrophy associated with increased levels of oxidative stress and impaired mitochondrial function [73]. In failing hearts, the dominant AMPKα isoform switches from AMPKα2 to AMPKα1, which accelerates the pathologic process. An overexpression of AMPKα2 in mouse hearts prevented the development of pressure overload-induced heart failure by increasing mitophagy and improving mitochondrial function. Phosphorylation of Ser495 in PINK1 by AMPKα2 is essential for efficient mitophagy to prevent the progression of heart failure. This increase in cardiac mitophagy is accompanied by the elimination of damaged mitochondria, an improvement in mitochondrial function, and a decrease in reactive oxygen species production and the apoptosis of cardiomyocytes [74].

## 6. Dilated Cardiomyopathy

Diverse alterations to the finely tuned mitochondrial dynamics mechanism can induce cardiac dysfunction. Drp1 knockout mice demonstrate neonatal lethality due to dilated cardiomyopathy, with severe mtDNA nucleoid clustering and deficiency of mitochondrial respiration [86]. Adult mice with inducible cardiac-specific deletion of Drp1 develop left ventricular dysfunction which is preceded by mitochondrial dysfunction [87]. A point mutation in mouse Drp1 in a highly conserved region of the M domain that alters intramolecular interactions within the Drp1 monomer reduces levels of mitochondria enzyme complexes and ATP depletion and also induces cardiomyopathy [88]. Conditional combined Mfn1/Mfn2 ablation in adult mouse hearts induced mitochondrial fragmentation, cardiomyocyte and mitochondrial respiratory dysfunction, and rapidly progressive and lethal dilated cardiomyopathy [89]. Thus, unopposed mitochondrial fission or fusion may both cause cardiac dysfunction, suggesting the critical importance of maintaining a well-balanced homeostasis of mitochondrial remodeling in the heart [87]. Pathogenic DNM1L mutations cause mitochondrial disorders with a highly variable clinical phenotype. A de novo DNM1L E410K mutation was associated with severe mitochondrial cardiomyopathy leading to nonischemic heart failure, cardiogenic shock and death. E410 is located in the domain of DNM1L that is important for tetramerization of the protein, and the substitution of a lysine at position 410 would thus be predicted to produce disease via a dominant-negative mechanism [90]. Fatal infantile mitochondrial encephalomyopathy, hypertrophic cardiomyopathy, and optic atrophy are associated with a homozygous gene mutation in OPA1, which results in a marked loss of steady-state levels of the native OPA1 protein, with a mitochondrial morphology consistent with abnormal mitochondrial membrane fusion [91].

The toxic cardiomyopathy induced by anthracycline anti-cancer drugs such as doxorubicin is mediated in part by increased mitochondrial fragmentation. Drp1-deficient mice are protected from doxorubicin-induced cardiac damage. Doxorubicin accelerates mitophagic flux, which was attenuated by Drp1 knockdown [49]. In addition, the inhibition of mitochondrial fission with Mdivi-1 protects the heart against doxorubicin-induced cardiac injury [50]. Knockdown of Parkin diminished doxorubicin-induced cell death, whereas an overexpression of Parkin had the opposite effect [49]. Doxorubicin induces mitophagy, increasing LC3, Beclin 1, decreasing p62, and co-localizing LC3 to mitochondria. Doxorubicin activates the PINK1/Parkin pathway and promotes translocation of PINK1/Parkin to mitochondria. In addition, doxorubicin inhibits the expression of PGC-1α and reduces the overall expression of mitochondrial proteins [75].

Mitophagy also plays an essential role in mitochondrial quality control in the heart during metabolic-induced cardiomyopathies, such as those induced by consumption of a high-fat diet. Impairment of mitophagy induces mitochondrial dysfunction and lipid accumulation, thereby exacerbating diabetic cardiomyopathy. Conversely, the activation of mitophagy protects against high-fat-diet-induced diabetic cardiomyopathy [76]. In cardiomyocytes, decreased OPA1 protein expression and increased O-GlcNAcylation of OPA1 by high glucose lead to mitochondrial dysfunction by increasing mitochondrial fragmentation, decreasing mitochondrial membrane potential, and attenuating the activity of mitochondrial complex IV. An overexpression of OPA1 attenuates high-glucose-induced mitochondrial fragmentation [77].

## 7. Pharmacological Modulation of Mitochondrial Dynamics and Quality Control

Therapeutic strategies to improve mitochondrial function in aging and cardiovascular diseases range from mitochondrial-targeted antioxidants, caloric restriction, caloric restriction mimetics, and exercise training [92]. Potential new strategies targeting mitochondrial dynamics have been explored, e.g., suppressing mitochondrial fission by chemical inhibitors has been shown to attenuate cardiac damage induced by ischemia-reperfusion, pressure overload, and doxorubicin [47,56,93].

The Drp1 inhibitor mitochondrial division inhibitor-1 (Mdivi-1) preserves mitochondrial morphology, reduces cytosolic calcium, and prevents cell death following ischemia-reperfusion in adult rat hearts [68]. Treatment of adult murine cardiomyocytes with Mdivi-1 reduces cell death and inhibits mitochondrial permeability transition pore opening after simulated ischemia-reperfusion injury, and in vivo treatment with Mdivi-1 reduces infarct size in mice subject to coronary artery occlusion and reperfusion [44]. A selective inhibitor of fission, P110, has been shown to inhibit the interaction of fission proteins Fis1/Drp1, decreasing mitochondrial fission, and improving bioenergetics in models of ischemia, including primary cardiomyocytes, ex vivo heart models, and in vivo myocardial infarction [80]. Dynasore is a cell-permeable small molecule that non-competitively inhibits the GTPase activity of dynamin1, dynamin2, and Drp1; prevents ischemia-reperfusion-induced elevation of left ventricular end diastolic pressure; decreases cardiac troponin I efflux during reperfusion; and reduces infarct size. In cardiomyocytes, Dynasore pretreatment prevents mitochondrial fragmentation induced by oxidative stress, reduces ATP depletion, and increases cell viability [94].

In addition to inhibiting mitochondrial fission, the promotion of mitochondrial fusion might be beneficial in the treatment of ischemia-reperfusion injury. The mitochondrial fusion promoter M1 is cardioprotective. When given before myocardial ischemia it increases mitochondrial fusion and improves mitochondrial function by decreasing ROS production, membrane depolarization, and mitochondrial swelling, resulting in decreased infarct size and improved cardiac function [69]. Melatonin has been shown to attenuate myocardial ischemia-reperfusion injury by improving mitochondrial fusion/mitophagy and activating the AMPK-OPA1 signaling pathways [81].

Spermidine treatment in mice promotes protective autophagy and mitophagy in cardiomyocytes and ameliorates hypertrophic remodeling of the aged heart, enhances diastolic function and extends the lifespan; however, its cardioprotective effects are due to several underlying mechanisms, including both direct cardiac and extracardiac effects [95]. Inhibitors of mTOR may also represent a strategy to induce general autophagy, and to protect against myocardial ischemia, cardiac hypertrophy, and cardiac aging. The statin simvastatin suppresses mTOR signaling and triggers the Parkin-dependent mitophagy required for cardioprotection [96,97].

The pharmacologic modulation of mitochondrial dynamics as a cardioprotective mechanism still remains a subject of intensive research. Given the fine balance between fission and fusion in maintaining mitochondrial homeostasis, and the constant changes in morphology and function in mitochondria, the impact of pharmacological interventions in acute vs. chronic settings deserves careful examination. Attention to off-target effects as well as an understanding of the limitations of these interventions are critical, as their beneficial effects are most likely going to be context-specific and perhaps limited in duration. Nonetheless, this area of research provides new and exciting pharmacological alternatives for manipulating and improving mitochondrial function.

## Figures and Tables

**Figure 1 biomedicines-10-01375-f001:**
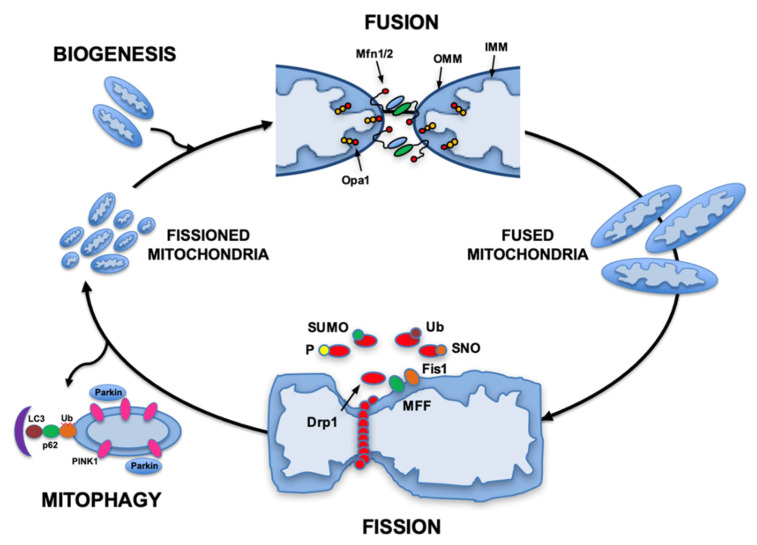
Mitochondrial dynamics and quality control. The mitochondrial remodeling processes of fission and fusion are both involved in regulating mitochondrial quality control. Mitochondrial fission is regulated by the interaction of mitochondrial proteins (Fis1 or MFF) with the cytosolic protein Drp1, which is translocated from the cytosol to the mitochondria fraction. This process is regulated by several different post-translational modifications (phosphorylation, sumoylation, s-nitrosylation and ubiquitination). During pathological stress, such as induced by ischemia-reperfusions, mitochondria with low membrane potential are marked for mitophagy by accumulation of Pink1, recruitment of Parkin, and are eliminated through autophagy. Mitochondrial fusion is regulated by the interaction of mitochondrial inner (Opa1) and outer membrane proteins (Mfn1/2). Mitochondrial biogenesis is the process that results in synthesis of new mitochondrial proteins and is regulated by the PPARγ coactivator transcriptional coactivators PGC-1α, PGC-1β, and PGC-related PRC. Abbreviations: Drp1, dynamin-related protein 1; Fis1, mitochondrial fission protein 1; IMM, inner mitochondrial membrane; LC3, Microtubule-associated proteins 1A/1B light chain 3B; MFF, mitochondrial fission factor; MFn, mitofusin; OMM, outer mitochondrial membrane; Opa1, optic atrophy 1; PGC-1, Peroxisome proliferator-activated receptor gamma coactivator 1; PINK1, PTEN-induced putative kinase 1; PPARγ, Peroxisome proliferator-activated receptor gamma; PRC, PGC-1-related coactivator; SNO, S-nitrosylated; SUMO, small ubiquitin-like modifier protein; Ub, ubiquitin. (From Gottlieb and Bernstein [3]. Based in part on Archer et al. [4]).

**Table 1 biomedicines-10-01375-t001:** Mitochondria Quality Control Alterations in Cardiovascular Diseases.

*Disease*	Mitochondrial Dynamics and Quality Control Alterations	Models	Phenotype
*Ischemic Heart Disease*	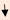 Mitophagy	Parkin knockout	More sensitive to myocardial infarction [43]
	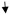 Mitophagy	Parkin overexpression	Protected against hypoxia-mediated cell death [43]
	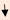 Mitophagy	PINK1 knockout	Larger myocardial infarcts than WT [62]
	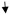 Mitophagy	PINK1 overexpression	Reduced cell death after simulated IR [62]
	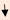 Mitophagy	Opa1 overexpression	Cardioprotective against hypoxia [63]
	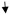 Mitophagy	Pigment epithelial-derived factor	Cardioprotective effects hypoxia [64]
	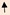 Mitophagy	Post-ischemic G protein-coupled estrogen receptor 1 activation	Cardioprotective effects against IR [65]
	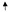 Mitophagy	ALDH2 activation with Alda-1	Cardioprotective against IR [66]
	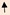 Mitophagy	Akap1 knockout	Larger infarct size, decreased survival [67]
	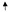 Fission	Neonatal murine cardiomyocytes and adult rat hearts after IR	Mitochondrial fragmentation and swelling within 30 min of IR [58,68]
	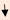 Fusion	Mitochondrial fusion promoter-M1 in IR	Cardioprotective against IR [69]
	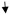 Fusion	OPA1 overexpression	Protected from IR [70]
*Cardiac Hypertrophy and Failure*	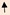 Mitophagy	Transverse aortic constriction	Mitophagy transiently activated at 3 to 7 days post transverse aortic constriction [71]
	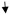 Mitophagy	Transverse aortic constriction	Mitophagy downregulated after 7 days post transverse aortic constriction [71]
	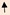 Fission	Stimulation of α1-adrenergic receptors with norepinephrine	Hypertrophy induced by norepinephrine with increased fission and decreased mitochondrial function [72]
	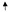 Fission	Dominant-negative Drp1	Prevented fission and blocked norepinephrine hypertrophic growth [72]
	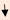 Mitophagy in heart failure	End-stage human heart failure	PINK1 protein levels are markedly reduced [73]
	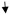 Mitophagy in heart failure	Samples from heart failure patients	Isoform shift from AMPKα2 to AMPKα1 in failing heart, decreased mitophagy and mitochondrial function [74]
	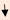 Mitophagy in heart failure	AMPKα2 overexpression	Increase in cardiac mitophagy, improvement in mitochondrial function [74]
*Cardiomyopathies*	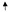 Fission	DRP1 knockout/doxorubicin	Doxorubicin accelerates mitophagy flux, attenuated by DRP1 knockdown [47]
	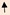 Fission	Isolated hearts/doxorubicin	Inhibition of mitochondrial fission with Mdivi-1 protects the heart against doxorubicin-induced cardiac injury [48]
	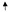 Mitophagy	Cells treated with doxorubicin	Doxorubicin induces mitophagy, activates the PINK1/Parkin pathway and inhibits the expression of PGC-1α [75]
	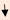 Mitophagy	High-fat diet induced diabetic cardiomyopathy	Activation of mitophagy protects against high fat induced diabetic cardiomyopathy [76]
	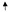 Fission	High glucose in neonatal cardiomyocytes	Decrease in mitochondrial membrane potential, overexpression of OPA1 attenuates mitochondrial fragmentation [77]

The arrow down—decreased, the arrow up—increased. Abbreviations: ALDH2: Aldehyde dehydrogenase2; Akap1: A-Kinase Anchoring Protein 1; AMPK: 5′-AMP-activated protein kinase; Drp1: dynamin-related protein 1; IR: Ischemia Reperfusion; Opa1: optic atrophy 1; PGC-1: Peroxisome proliferator-activated receptor gamma coactivator 1; PINK1: PTEN-induced putative kinase 1.

## Data Availability

Not applicable.

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
