# Peer review of "Mitochondrial Quality Control in the Heart: The Balance between Physiological and Pathological Stress"

_biomedicines, 2022, doi:10.3390/biomedicines10061375_

Round 1
Reviewer 1 Report
The review work presented by Giovanni Fajardo and colleagues is well written, clear, and easy to read. As conceived is an outstanding review. The topic is interesting and therefore, it adds clustered information to the subject area of mitochondrial dynamics in the cardiac tissue linked to acute and chronic exercise as well as cardiac pathophysiology and diseases. The author performed a very well-conceived overview of the role of mitochondrial dynamics in the cardiac tissue remodeling that still remains a subject of intensive research as a cardio-protective mechanism and molecules that have an impact on these biochemical mechanisms are certain a cutting edge pharmacological treatments.
Author Response
We are appreciate the positive comments to our review and we hope that it can contribute to a better understanding of the role of mitochondrial dynamics and quality control in physiological and pathological adaptations to cardiac stress.
Reviewer 2 Report
Authors focused on pathologic and physiologic cardiovascular stress in context of mitochondrial function and quality control. Adaptation of heart muscle mitochondria to physiological processes such as acute and chronic exercise and pathological adaptations including ischemic heart disease, cardiac hypertrophy and failure and cardiomyopathy are discussed. The last part is devoted to pharmacological modulation of mitochondrial quality control.
Comments:
Such a general remark - fusion and fission are mitochondrial dynamics and affect mitochondrial quality control. But mitochondrial dynamics and mitochondria quality control are still two different things.
The text is not very readable and basically a lot of information that is mentioned in the text is then summarized in Table 1.
An overview of the various pathological situations on the models would certainly deserve further mention of examples from human medicine- e.g. cardiac impairment in patient with DNM1L mutation etc.
Page 2 line 58 - From the current sentence formation, it might seem that mitophagy is dependent on Drp1. There are many publications showing the independence of mitophagy from Drp1-mediated division of mitochondria.
Page 2 lines 69 - p62 is not the only adapter protein recruited. In addition, it is not a primary adapter protein, so it is not essential for the course of mitophagy, see:
Lazarou, M., Sliter, D.A., Kane, L.A., Sarraf, S.A., Wang, C., Burman, J.L., Sideris, D.P., Fogel, A.I., Youle, R.J., 2015. The ubiquitin kinase PINK1 recruits autophagy receptors to induce mitophagy. Nature 524, 309–314. https://doi.org/10.1038/nature14893
It would be better to write that adapter proteins such as xxx are recruited…
Throughout the article, authors focus on PINK1/Parkin mitophagy, which plays a major role, but as far as receptor-mediated mitophagy is concerned, it is activated especially in hypoxia. It would therefore be worth considering whether to add comments on receptor-mediated mitophagy (BNIP3 and FUNDC1) throughout the review.
Table 1.
It would be appropriate to add to each piece of information the relevant citation.
Figure 1.
Abbreviation labels are very small (LC3, PINK) in the figure.
It is necessary to write full name of abbreviations in the caption of the image.
What is MFF in figure..? In description is MPP mentioned….
Round 2
Reviewer 2 Report
Accept in present form.